# AutoVSR: Automatic Visual-to-Symbolic Reasoning for Symbolic Expression Generation from Circuit Schematic

**Zhe Xiao** [* 1]  **Longfei Li** [* 1]  **Xu He** [† 1]  **Haoying Wu** [2]  **Zixing Zhang** [1]  **Mingyu Liu** [3]

## Abstract

Symbolic expressions can effectively characterize and predict circuit behavior, but deriving them directly from circuit schematics is challenging. This process requires accurate visual-to-symbolic construction of circuit structure from images and correct multi-step symbolic derivation, both of which impose strict correctness requirements. This work proposes AutoVSR, an automated framework for visual-to-symbolic generation of circuit expressions using Vision Language Models (VLMs). By reconstructing circuit diagrams into an executable intermediate representation (Executable IR) and leveraging a symbolic solver for reasoning, AutoVSR significantly improves the accuracy of symbolic expression generation. AutoVSR introduces two key innovations: an IR construction method guided by component rule retrieval and verification-based feedback, and a symbolic solver implemented as a planning agent equipped with a symbolic tool library for reliable multi-step derivation. Compared with end-to-end VLM approaches and specialized methods on the main symbolic expression generation task, AutoVSR achieves accuracy improvements of 30.01–59.45% and 41.96–51.84%, respectively. Moreover, AutoVSR surpasses closed-source state-of-the-art VLMs in inference cost and computational efficiency. Code is available at https://github.com/LongfeiLi1/AutoVSR.

## 1. Introduction

Integrated circuits form the physical foundation of the modern information society, enabling key functions such as

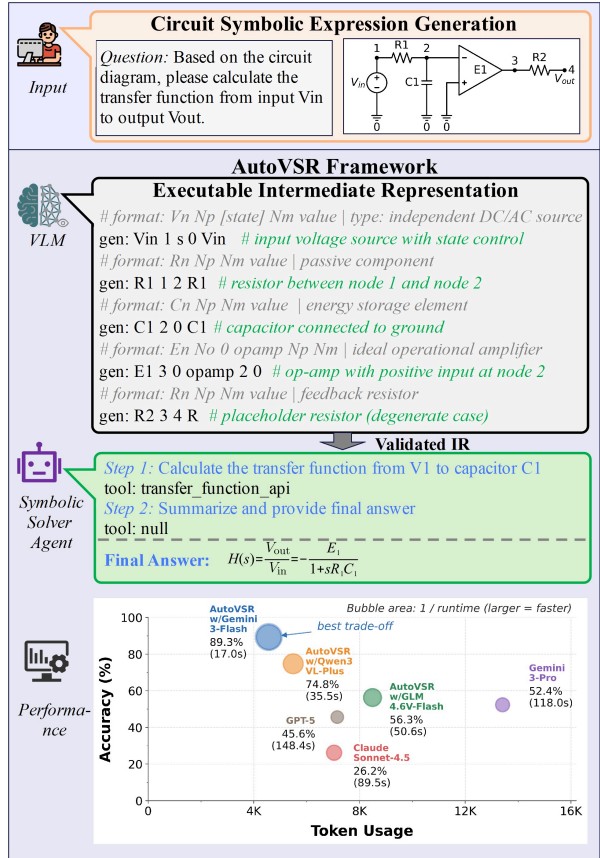

*Figure 1.* AutoVSR Framework Workflow & Performance Results

information transmission, signal processing, and power management in electronic systems (Hao et al., 2021). To meet performance requirements such as stability and dynamic response, engineers often rely on symbolic analysis to characterize and predict circuit behavior during the design process (Gielen et al., 1994). A key step in symbolic analysis is to convert the structural information in visual circuit schematics into computable symbolic expressions that describe circuit behavior, such as input–output relationships and related analytical forms including transfer functions (Gielen & Sansen, 1991). In practice, obtaining these expressions requires interpreting schematic images to identify components and recover connectivity, followed by step-by-step symbolic derivation and simplification. However, this expertise-dependent workflow is labor-intensive. There-

---

[*]Equal contribution. [†] Xu He is the corresponding author, email: dawn.hx@gmail.com. [1]Hunan University, Changsha, China [2]Wuhan University of Technology, Wuhan, China [3]Huazhong University of Science and Technology, Wuhan, China.

fore, an automated visual-to-symbolic reasoning framework is needed to bridge schematic images and symbolic expressions, improving efficiency and shortening design cycles.

The emergence of multimodal large language models (MLLMs) has brought new opportunities to circuit-related tasks (Chen et al., 2024). Current work mainly falls into two categories. **(1) Circuit topology generation and reconstruction.** These works focus on inferring the connectivity among circuit components, such as recovering netlists from schematics (Huang et al., 2025; Bhandari et al., 2025; Kulkarni et al., 2025) or synthesizing new circuit topologies (Vijayaraghavan et al., 2025; Gao et al., 2025). **(2) Assisted circuit sizing and optimization.** Under a fixed topology, these works optimize device sizes and bias parameters (Yin et al., 2025; Karthik Somayaji & Li, 2025), typically by leveraging MLLMs to guide parameter search and optimization (Shi et al., 2025; Liu et al., 2025) with numerical simulation in the loop (Kochar et al., 2025; Shen et al., 2026). However, both directions are formulated differently from *symbolic expression generation* setting. They primarily operate on circuit structures or parameters and rely on SPICE-based (Anderson et al., 2016) numerical simulation for evaluation, whereas our goal is to generate computable and verifiable target symbolic expressions that characterize circuit behavior directly from circuit schematic images via visual-to-symbolic reasoning.

To the best of our knowledge, only one closely related work (Akbari et al., 2026) has explored *symbolic expression generation* from circuit images. It employs Chain-of-Thought (CoT) (Wei et al., 2022) prompting to guide vision-language models (VLMs) through an end-to-end generation process, including component identification, modeling of circuit elements, and symbolic derivations based on nodal or loop-style reasoning, ultimately producing target symbolic expressions. However, both the intermediate reasoning and the final expressions are still produced directly by the VLM in an end-to-end manner, without explicit, step-by-step verification of intermediate states. Achieving truly reliable visual-to-symbolic *symbolic expression generation* still faces two key challenges. **(a) Reliable construction of symbolic representations from visual inputs (Lu et al., 2021).** Symbolic expression generation relies on accurate structured representations, while circuit schematics are given as images and do not directly provide machine-readable symbols such as device types, connectivity, or variable definitions. Errors in this visual-to-symbolic construction stage can change the circuit model and propagate to subsequent derivations. **(b) Correctness of multi-step symbolic derivation (Pan et al., 2025).** Target symbolic expressions typically require a sequence of dependent derivation and simplification steps. The derivation is sensitive to small errors in equation setup, sign conventions, and algebraic manipulation, and such errors can propagate across steps, leading to incorrect final expressions.

From the perspective of symbolic computation, toolchains such as Lcapy and SymPy can support circuit-level symbolic analysis and algebraic manipulation, but they operate on structured symbolic inputs rather than directly processing schematic images. Therefore, they do not by themselves address the visual-to-symbolic construction required for schematic-based symbolic expression generation.

This work proposes AutoVSR, a vision-to-symbolic reasoning framework based on VLMs for symbolic expression generation from circuit schematics, as shown in Fig. 1. To ensure reliable symbolic construction from visual inputs, AutoVSR introduces an executable intermediate representation (Executable IR) that explicitly encodes device types and connectivity and can be directly instantiated as a circuit object by a symbolic engine. To improve the accuracy of IR construction, AutoVSR further employs dynamic context prompting and verification-based iterative feedback to enforce physical consistency and reduce visual-to-symbolic errors. To guarantee accurate in multi-step symbolic derivation, AutoVSR adopts a planning–execution decoupling strategy. The VLM handles high-level reasoning by decomposing analysis goals into explicit steps, while dedicated symbolic tools execute equation construction and algebraic computation. This ensures rule-consistent derivations with clear intermediate results and avoids error accumulation. By integrating visual perception, executable symbolic representation, and tool-driven derivation, AutoVSR enables accurate visual-to-symbolic expression generation. Our key contributions are summarized as follows:

- To enable automated visual-to-symbolic reasoning for schematic, we propose AutoVSR and introduce an Executable IR that converts circuit schematics into analyzable symbolic circuit objects.
- To make schematic understanding robust, we design a visual perception mechanism based on dynamic context prompting and dual-ended verification feedback, reducing topological and syntactic errors in circuit structure generation.
- To reduce error propagation in multi-step symbolic derivation, we develop a symbolic tool-augmented solver that decouples high-level reasoning from execution, delegating equation construction and computation to symbolic tools.
- Experiments across five circuit types show that AutoVSR improves accuracy on the main symbolic expression generation task by **30.01–59.45%** over end-to-end VLM baselines and by **41.96–51.84%** over the specialized baseline. In addition, against stronger closed-source models, AutoVSR remains accuracy-competitive while achieving lower inference cost and higher efficiency.

## 2. Preliminaries and Related Works

**Symbolic Circuit Expressions and Toolchains.** In circuit design and analysis, many key quantities can be represented as symbolic expressions, such as node voltages, branch currents, and input to output relations. A common approach is to formalize circuit topology together with element constitutive relations into a system of equations using Kirchhoff laws (Kirchhoff, 1845), and to obtain target expressions through symbolic solving and algebraic simplification (Ho et al., 1975; Gielen & Sansen, 1991). Since symbolic expressions may suffer from expression swell on larger or more complex circuits, practical symbolic analyzers often incorporate simplification, factorization, and approximation or term pruning strategies to improve usability (Fernández et al., 1994). Existing open source symbolic toolchains provide reusable back end support for this workflow. SymPy (Meurer et al., 2017) offers general purpose symbolic manipulation, matrix solving, and simplification utilities. Lcapy (Hayes, 2022), built on SymPy, further supports circuit oriented inputs such as SPICE-like netlists and port compositions, and can automatically derive impedances, input to output expressions, and related responses. However, despite their effectiveness, they require a structured symbolic circuit description as input, and a visual schematic does not directly provide such machine readable symbols. Therefore, these toolchains cannot be directly applied to circuit images without first manually converting the visual content into the circuit representation required by these tools as input.

**MLLMs for Circuit Tasks.** MLLMs have been increasingly applied to a variety of circuit-related tasks. A recurring workflow is to translate a schematic or description into a machine tractable intermediate representation, such as an explicit connectivity description, a graph, or a sequence, so that component types and net relations become amenable to learning based inference (Gao et al., 2025). Recent systems use MLLMs to complete or edit circuit structure, ranging from schematic to netlist recovery (Huang et al., 2025; Bhandari et al., 2025; Kulkarni et al., 2025) to topology proposal and exploration via generative search or reinforcement learning (Vijayaraghavan et al., 2025; Lai et al., 2025; Chang et al., 2025). In addition, MLLMs have been incorporated into sizing workflows as proposal modules for device dimensions and bias choices (Yin et al., 2025; Karthik Somayaji & Li, 2025), where the model suggests candidate parameter sets (Shi et al., 2025; Liu et al., 2025) and an outer loop iteratively evaluates and refines them using simulation feedback to satisfy performance constraints (Kochar et al., 2025; Shen et al., 2026). Across these lines of work, the formulation is typically structure or parameter centric, where circuit behavior is evaluated by SPICE simulation and the model is guided by numerical feedback. In contrast, our work is expression centric and targets generating explicit symbolic circuit expressions from schematic visuals. Such expressions make circuit behavior explicit in symbolic form, clarifying the physical mechanisms and enabling systematic reasoning through exact algebraic manipulation and simplification.

## 3. Methodology

### 3.1. Method Overview

AutoVSR is a multimodal agent designed for complex circuit symbolic reasoning by transforming visual inputs and task queries into executable symbolic representations. Unlike traditional end to end approaches, AutoVSR adopts a structured framework that decouples visual to symbolic reconstruction from symbolic reasoning. As shown in Fig. 2, the framework consists of two main stages, visual to textual topology generation and IR driven symbolic reasoning. In the first stage, AutoVSR constructs an Executable IR that explicitly encodes device semantics and circuit connectivity. Guided by a task router and component detection module, the system retrieves corresponding component rules to form dynamic context prompting, which directly steers IR generation and suppresses irrelevant visual noise. To ensure reliability, AutoVSR incorporates verification based iterative feedback that enforces physical and structural consistency, enabling the model to automatically correct visual to symbolic errors and produce a validated IR. In the second stage, AutoVSR performs symbolic reasoning on the validated IR. Under a planning–execution decoupling strategy, the VLM decomposes the analysis objective into explicit steps, while symbolic tools carry out the corresponding computations. By relying on deterministic solvers, AutoVSR mitigates error accumulation and produces rule-consistent reasoning with transparent intermediate results.

### 3.2. Visual-to-Textual Topology Generation

Since directly applying native VLMs to circuit reasoning often mixes visual perception with symbolic inference, AutoVSR first focuses on constructing an explicit and structured circuit intermediate representation as a reliable foundation for reasoning. This representation is crucial because its fidelity and completeness directly determine the effectiveness of subsequent symbolic analysis.

To enable accurate IR construction from unstructured multimodal inputs, AutoVSR incorporates a dynamic context prompting, together with a verification enhanced feedback flow, to improve generation reliability. Specifically, the dynamic context injection mechanism includes IR structure selection, task classification and analysis, and component detection with rule retrieval, while the verification enhanced feedback flow enforces physical and structural consistency through iterative validation.

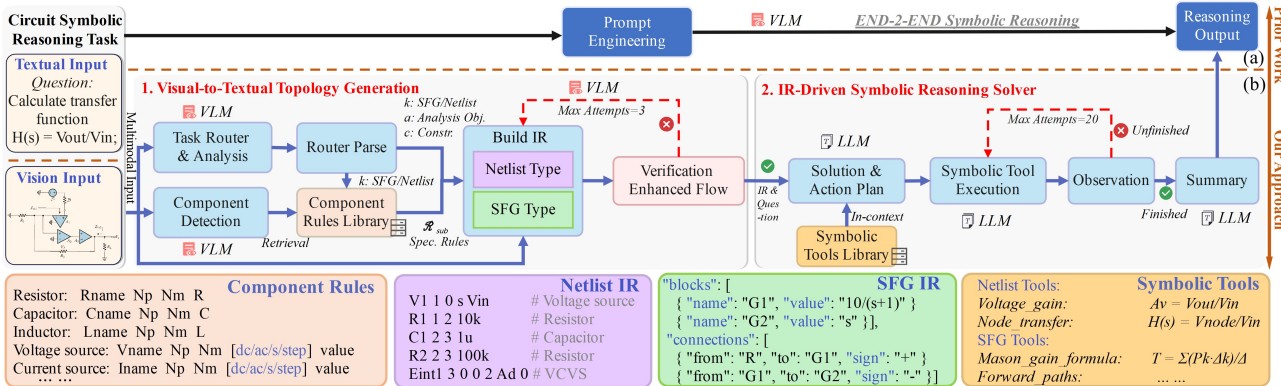

*Figure 2.* **Method Overview**. (a) **Prior Method.** Direct end-to-end inference where the VLM takes the circuit image and symbolic question as input to directly generate the answer. (b) **Our Approach.** A verification-enhanced framework that converts multimodal inputs into verifiable IR. It utilizes an automated error-correction mechanism for IR generation, followed by LLM-based dynamic tool planning and symbolic derivation for precise reasoning.

### 3.2.1. IR STRUCTURE SELECTION

Considering that different types of circuits require different representations for effective symbolic analysis, we adopt two complementary intermediate representations tailored to the underlying symbolic solvers. As illustrated in Fig. 3, schematic level circuits are represented using a netlist based circuit topology representation (**Netlist IR**) in a SPICE like format, which serves as the native input format of the Lcapy symbolic circuit analysis library and enables direct symbolic computation on circuit topology. In contrast, system level circuits are represented using a signal flow graph based representation (**SFG IR**) in a JSON structured format, which is specifically designed to interface with our Mason (Mason, 1953) based symbolic solver for handling directed signal flows and feedback loops. By selecting the appropriate representation according to circuit type and analysis granularity, AutoVSR maps each task to an executable intermediate representation that is directly compatible with the corresponding symbolic reasoning engine.

### 3.2.2. TASK ROUTER AND ANALYSIS

Due to the fundamentally different construction objectives of the SFG and Netlist IR, system level analysis requires directed signal flow and gain modeling, whereas schematic level analysis focuses on component connectivity. Accordingly, the SFG and Netlist follow distinct constraint spaces, namely directed topological constraints $\mathcal{C}_{\text{sfg}}$ and component level syntactic constraints $\mathcal{C}_{\text{net}}$. Without explicit separation, direct generation forces the model to optimize over the conflicting union $\mathcal{C}_{\text{sfg}} \cup \mathcal{C}_{\text{net}}$, resulting in format confusion and semantic interference. To avoid this issue, AutoVSR explicitly selects the appropriate IR and constraint space prior to generation. Inspired by the Mixture of Experts (MoE) (Shazeer et al., 2017), we leverage multimodal inputs, namely the circuit image $I$ and the textual query $Q$, to define a gating function $k = g(I, Q) \in \{\text{sfg}, \text{netlist}\}$

that selects the target IR space. This routing mechanism confines generation to a single constraint space, eliminating task ambiguity. In addition to routing, the module extracts a meta constraint tuple $(a, c) = h(I, Q)$, where $a$ specifies the analysis objective and $c$ encodes explicit constraints such as I O nodes and symbolic parameters. The module outputs a structured instruction triplet $\mathcal{S} = (k, a, c)$, which serves as the core state for subsequent generation.

### 3.2.3. COMPONENT DETECTION AND RULES RETRIEVAL

Although the instruction triplet $\mathcal{S}$ specifies the high-level generation objective, translating it into solver-compatible intermediate representations remains challenging due to strict component-level syntactic constraints. Symbolic solvers such as Lcapy require distinct syntax for different components, which cannot be resolved by high-level instructions alone, while injecting the full rule library introduces excessive context noise. To address this issue, we adopt a vision-based dynamic rule retrieval strategy. The VLM first detects the set of components $\mathcal{T}$ present in the circuit image $I$:

$$\mathcal{T} = \text{Detect}(I). \tag{1}$$

Based on the target format $k$ and the detected components $\mathcal{T}$, the system selectively retrieves the relevant subset of syntax rules:

$$\mathcal{R}_{sub} = \{\text{Rule}(t, k) \mid t \in \mathcal{T}\}. \tag{2}$$

The retrieved rules are combined with the instruction triplet to form a focused context $\mathbf{C}$, which guides IR generation:

$$\mathbf{C} = \mathcal{S} \oplus \mathcal{R}_{sub}, \quad IR = \text{VLM}(I, Q; \mathbf{C}). \tag{3}$$

This design restricts generation to syntax relevant to the active components, effectively eliminating irrelevant rules and improving IR correctness. Detailed component specifications are provided in Appendix A.5.

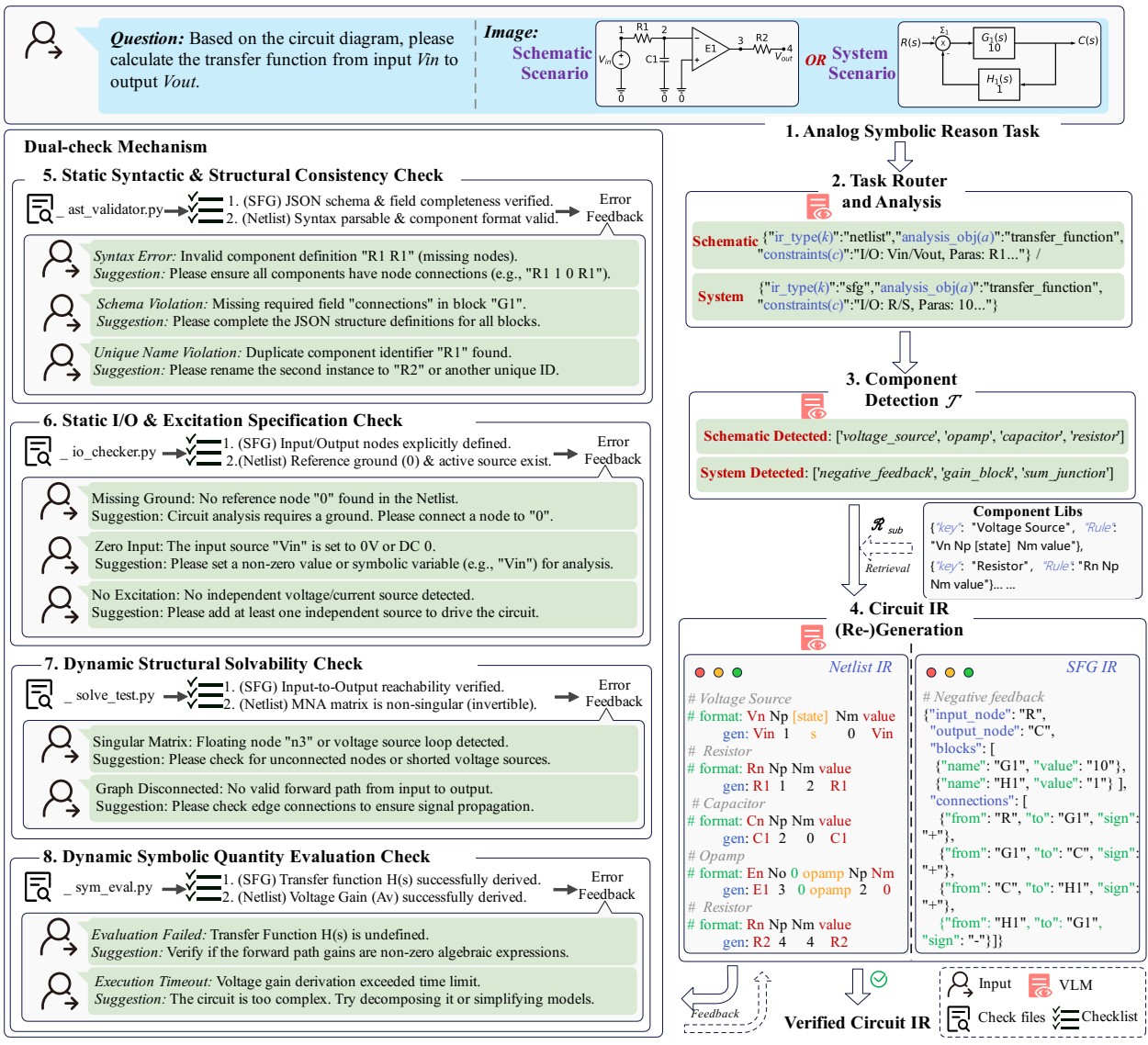

*Figure 3.* **Visual-to-Textual Topology Generation Workflow.** Including schematic and query input, task routing and analysis, component detection with rule retrieval, and executable IR generation with iterative verification feedback.

### 3.2.4. VERIFICATION ENHANCED FEEDBACK FLOW

Various errors ranging from syntax violations to topological inconsistencies are often observed in IR generated by VLMs. Consequently, guiding VLMs to correct these IRs based on error feedback is crucial. Numerous studies (Lai et al., 2025; Olausson et al., 2024) have suggested that providing LLMs with relevant error information significantly facilitates error correction. However, for symbolic circuit reasoning, ensuring validity requires verification beyond basic format compliance, extending to physical solvability and graph-theoretic executability. To address potential hallucinations, we implement a dual-check mechanism combining static AST parsing and dynamic simulation, dividing the flow into four ordered stages as illustrated in Fig. 3: (1) static syntactic and structural consistency check, (2) static I/O and excitation specification check, (3) dynamic structural solvability check, and (4) dynamic symbolic quantity evaluation check. Initiating the flow, the static syntactic check utilizes AST-based parsers to validate IR well-formedness, rigorously scrutinizing component attributes element-by-element against predefined grammar specifications. Subsequently, the static I/O check verifies minimal physical preconditions, confirming the existence of reference grounds and valid independent sources for Netlists, while ensuring explicit node definitions for SFGs. Transitioning to dynamic verification, the structural solvability check executes backend solvers to detect topological singularities (e.g., shorted sources, floating nodes) while validating Modified Nodal Analysis (MNA) (Ho et al., 1975) matrix invertibility for Netlists and input-to-output reachability for SFG. Finally, the symbolic

quantity evaluation check attempts to derive target expressions like transfer functions, confirming the mathematical robustness of the IR. For any errors, structured feedback is returned to the VLM for regeneration, allowing up to three generations to mitigate self-correction limitations.

### 3.3. IR-Based Symbolic Reasoning Solver

Studies (Wu et al., 2024; Dziri et al., 2023) show that LLMs suffer from error accumulation in long-chain mathematical reasoning, especially when required to perform precise symbolic computation across multiple steps. To mitigate this limitation, we decouple logical reasoning from mathematical execution. After the intermediate representation is verified, the LLM focuses on high-level planning, while deterministic algorithmic tools handle symbolic operations, ensuring correctness throughout the reasoning process.

**Plan-and-Execute Symbolic Reasoning Agent.** Inspired by strategy decomposition (Zhou et al., 2023) and tool learning (Qin et al., 2024), our solver implements a closed-loop "Plan-and-Execute" workflow comprising: (1) circuit instantiation, (2) strategic planning, (3) tool-oriented execution, and (4) summary. Initially, we instantiate the verified IR into backend objects to establish a physically grounded state space. To mitigate computational hallucinations, the LLM decomposes the query into an atomic execution blueprint, driving an "Action-Observation" loop that refines reasoning via structured tool feedback. Finally, the Summary stage synthesizes the cumulative reasoning history into a conclusive symbolic output. To guarantee termination, we enforce a strict 20-step budget that compels the agent to finalize its answer if the limit is reached, effectively preventing infinite logical loops.

**Circuit Tools Library.** To support granular tool invocations by the agent, we constructed a library of 28 atomic APIs that encapsulate symbolic solvers into dynamic interfaces. Depending on the IR, the system employs the SFG toolkit centered on the Mason rule for graph-theoretic path enumeration and transfer derivation, or the Netlist toolkit leveraging Lcapy for precise voltage, impedance, and feedback probing. Crucially, these solvers are encapsulated within sandboxed, timeout-protected subprocesses, ensuring system stability even when handling computationally expensive symbolic operations. Details are provided in Appendix A.4.

## 4. Experimental Results

To address the difficulty of generating correct symbolic expressions directly from visual schematics, where crucial circuit semantics are implicit and require multi-step derivation, we conduct experiments to compare AutoVSR with end-to-end VLM baselines, specialized approaches, and closed-source proprietary reasoning models on a benchmark spanning five circuit types. Experiments are conducted on one node with two Intel 8-core Xeon Silver 4215R CPUs, 256GB main memory. Since several compared models are only accessible through remote services, all models are evaluated via API-based inference to ensure consistent evaluation.

**Benchmark.** To comprehensively evaluate AutoVSR on symbolic expression generation from circuit schematics, we select all symbolic-expression-generation tasks from CircuitSense (Akbari et al., 2026), including transfer function generation and transient response expression generation. Both selected tasks require understanding circuit structures and performing symbolic modeling and derivation, making them representative benchmarks for symbolic circuit expression generation. Since some samples may be subject to answer memorization by pretrained models, as noted in CircuitSense, we filter the benchmark accordingly and correct inaccurate annotations to ensure reliable evaluation. The final benchmark contains 5020 samples, including 1376 transfer function generation samples and 3644 transient response expression generation samples. It is divided into five types, including `Type1` with 1146 samples of pure resistive networks, `Type2` with 2671 samples of RLC circuits, `Type3` with 464 samples of small-signal models, `Type4` with 511 samples of module-level circuits, and `Type5` with 228 samples of system-level block diagrams.

**Evaluation.** Since algebraically equivalent expressions may take different forms after expansion or simplification, evaluation is based on symbolic equivalence rather than string matching. Therefore, we follow the evaluation protocol of CircuitSense (Akbari et al., 2026), using SymPy for mathematical equivalence verification, with numerical substitution at multiple test points as a fallback when symbolic verification fails.

**Hyper-parameters.** To ensure fair and consistent evaluation, all compared VLMs are tested under identical hyper-parameter settings unless otherwise specified. The `temperature` of all VLMs is set to 1.0, `max_tokens` is limited to 4096, and the `reasoning_mode` is set to minimal throughout the experiments. For AutoVSR-specific settings, the IR construction stage allows up to three rounds of self-correction based on solver feedback to handle potential construction errors. The symbolic solver is capped at 20 execution steps per problem to avoid infinite loops. Lcapy is used as the underlying symbolic execution engine.

### 4.1. Performance Analysis

To evaluate the performance of our AutoVSR, we conduct a systematic comparison against two distinct categories of baselines: (1) End-to-End VLMs: We select a comprehensive set of mainstream VLMs, encompassing both proprietary and open-source leaders. These include

*Table 1.* **Performance Comparison of AutoVSR and End-to-End VLMs on Transfer Function Generation.** We apply AutoVSR as a unified enhancement framework to native VLMs, including four closed source models, Gemini-3-Flash, GPT-5-mini, Qwen3-VL-Plus, and Claude-Haiku-4.5, as well as three lightweight open source models, Qwen3-VL-32B-Instruct, Llama-4-17B-128E, and GLM-4.6V-Flash. Overall, AutoVSR consistently outperforms these native models across all evaluated settings.

| Model Type | Method | Type 1 | | Type 2 | | Type 3 | | Type 4 | | Type 5 | | Overall | |
|---|---|---|---|---|---|---|---|---|---|---|---|---|---|
| | | Acc. (%) | Imp. ↑ | Acc. (%) | Imp. ↑ | Acc. (%) | Imp. ↑ | Acc. (%) | Imp. ↑ | Acc. (%) | Imp. ↑ | Acc. (%) | Imp. ↑ |
| Closed Source | Gemini-3-Flash | 37.95 | +46.99 | 35.67 | +53.51 | 23.06 | +51.29 | 11.93 | +72.73 | 3.95 | +83.77 | 23.40 | +59.45 |
| | AutoVSR(Gemini-3-Flash) | **84.94** | | **89.18** | | **74.35** | | **84.66** | | **87.72** | | **82.85** | |
| | GPT-5-mini | 9.04 | +83.13 | 9.36 | +68.13 | 7.97 | +23.50 | 13.64 | +40.34 | 16.23 | +14.91 | 10.54 | +42.51 |
| | AutoVSR(GPT-5-mini) | **92.17** | | **77.49** | | **31.47** | | **53.98** | | **31.14** | | **53.05** | |
| | Qwen3-VL-Plus | 18.07 | +74.70 | 19.30 | +74.85 | 13.36 | +35.78 | 19.89 | +50.00 | 15.79 | +32.02 | 16.64 | +51.38 |
| | AutoVSR(Qwen3-VL-Plus) | **92.77** | | **94.15** | | **49.14** | | **69.89** | | **47.81** | | **68.02** | |
| | Claude-Haiku-4.5 | 9.64 | +64.46 | 10.53 | +68.71 | 10.56 | +18.10 | 9.09 | +15.91 | 5.70 | +14.04 | 9.67 | +35.10 |
| | AutoVSR(Claude-Haiku-4.5) | **74.10** | | **79.24** | | **28.66** | | **25.00** | | **19.74** | | **44.77** | |
| Open Source | Qwen3-VL-32B-Instruct | 15.06 | +78.31 | 13.74 | +75.73 | 7.11 | +28.67 | 11.93 | +58.52 | 17.98 | +25.44 | 12.14 | +49.63 |
| | AutoVSR(Qwen3-VL-32B-Instruct) | **93.37** | | **89.47** | | **35.78** | | **70.45** | | **43.42** | | **61.77** | |
| | Llama-4-17B-128E | 10.84 | +42.77 | 10.23 | +45.33 | 6.47 | +18.53 | 7.39 | +28.41 | 13.16 | +22.37 | 9.16 | +30.01 |
| | AutoVSR(Llama-4-17B-128E) | **53.61** | | **55.56** | | **25.00** | | **35.80** | | **35.53** | | **39.17** | |
| | GLM-4.6V-Flash | 3.61 | +74.70 | 8.19 | +65.20 | 6.25 | +30.17 | 8.52 | +34.09 | 10.09 | +16.66 | 7.34 | +42.51 |
| | AutoVSR(GLM-4.6V-Flash) | **78.31** | | **73.39** | | **36.42** | | **42.61** | | **26.75** | | **49.85** | |
| Avg. | Base Model | 14.89 | +66.43 | 15.29 | +64.49 | 10.68 | +29.44 | 11.77 | +42.86 | 11.84 | +29.89 | 12.70 | +44.37 |
| | AutoVSR | **81.32** | | **79.78** | | **40.12** | | **54.63** | | **41.73** | | **57.07** | |

Gemini-3-Flash (Doshi, 2025), GPT-5-mini (Singh et al., 2025), Claude-Haiku-4.5 (Anthropic, 2025a), Qwen3-VL-Plus (Bai et al., 2025), Qwen3-VL-32B-Instruct (Bai et al., 2025), Llama-4-17B-128E (Llama, 2025), and GLM-4.6V-Flash (Team et al., 2025). (2) Circuit-Specific Method: We also compare against CircuitSense (Akbari et al., 2026), a specialized baseline that employs COT to guide the analysis. It explicitly identifies components and constructs impedance models, subsequently applying nodal or loop analysis to derive symbolic expressions.

**Comparison with End-to-End VLM Baselines on Transfer Function Generation.** Table 1 compares the performance of the proposed AutoVSR with the end-to-end baseline across seven VLMs. The first row represents the accuracy on different circuit types, along with the percentage accuracy improvement of our method over other VLMs. The second column represents the performance of end-to-end VLMs with their AutoVSR-enhanced counterparts. Overall, AutoVSR achieves an accuracy improvement ranging from 30.01% to 59.45%. Notably, across different task types, Type1 and Type2 exhibit the largest improvements, with gains of 66.43% and 64.49%, respectively. This is because Type1 and Type2 consist of relatively simple circuit structures. With dynamic context prompting, AutoVSR can more accurately translate circuit structures into Executable IR, whereas end-to-end models often overlook fine-grained component details, such as explicit component connections and parameter roles, leading to degraded reasoning performance. For more complex circuit types, Au-

toVSR achieves performance improvements of 29.44% on Type3 and 42.86% on Type4. These gains arise because Type3 involves dependent sources that are highly sensitive to terminal correspondence and polarity, while Type4 includes functional modules such as op-amps requiring precise port semantics and connection topology. End-to-end VLMs often suffer from local visual misinterpretations in these cases, which propagate into systematic errors during symbolic derivation. In contrast, AutoVSR enforces explicit structural constraints through IR-level reasoning and verification feedback, effectively preventing local recognition errors from escalating into global reasoning failures.

For Type5, the task is inherently more challenging because it focuses on system-level signal flow rather than component-level schematics. As a result, base models achieve an average accuracy of only 11.84%, while AutoVSR attains an absolute improvement of 29.89%. This task requires deriving the overall transfer function from signal flow graphs with complex and deeply nested feedback structures. It involves enumerating all forward paths and feedback loops, identifying non-touching loops, and computing their respective gains before applying Mason's Gain Formula, which demands global graph-level reasoning. End-to-end VLMs struggle with this combinatorial, multi-step symbolic reasoning process. In contrast, AutoVSR explicitly implements these procedures through a symbolic reasoning tool library, enabling accurate and reliable system-level analysis. In summary, compared with end to end methods, AutoVSR benefits from an Executable IR, which ensures reliable vi-

sual construction, and from a tool enhanced solver, which addresses the accuracy and interpretability challenges in symbolic derivation.

*Table 2.* **Performance Comparison of AutoVSR and End-to-End VLMs on Transient Response Expression Generation.** AutoVSR substantially improves overall transient response expression generation accuracy over native end-to-end VLM baselines.

| Method | Acc. (%) | Imp. ↑ |
|---|---|---|
| Gemini-3-Flash | 2.85 | +82.99 |
| AutoVSR(Gemini-3-Flash) | **85.84** | |
| GLM-4.6V-Flash | 0.69 | +62.56 |
| AutoVSR(GLM-4.6V-Flash) | **63.25** | |

**Generalization Evaluation on Transient Response Expression Generation.** To verify the generalization ability of AutoVSR, we further evaluate it on transient response expression generation. Different from transfer function generation, which focuses on frequency-domain input–output relationships, transient response generation requires time-domain reasoning over initial/final conditions and dynamic circuit behavior. As shown in Table 2, end-to-end VLMs achieve only 2.85% and 0.69% accuracy with Gemini-3-Flash and GLM-4.6V-Flash, respectively. In contrast, AutoVSR improves the accuracy to 85.84% and 63.25%, yielding absolute gains of 82.99 and 62.56 percentage points. These results show that AutoVSR generalizes well across symbolic generation tasks because it separates visual structure extraction from symbolic reasoning. The Executable IR provides a task-agnostic structural representation, while the tool-enhanced solver adapts the reasoning process to different symbolic generation objectives.

*Table 3.* **Performance Comparison between AutoVSR and a Circuit-Specific Method across Different Models.** We compare AutoVSR with CircuitSense (Akbari et al., 2026), a circuit specific chain of thought method with injected domain knowledge, across different tasks and models. AutoVSR consistently outperforms CircuitSense in overall performance.

| Task | Method | Model | Acc. (%) | Imp.↑ |
|---|---|---|---|---|
| Transfer Function | CircuitSense | Gemini-3-Flash | 40.89 | - |
| | | Qwen3-VL-Plus | 16.18 | - |
| | | GLM-4.6V-Flash | 7.80 | - |
| | AutoVSR | Gemini-3-Flash | **82.85** | 41.96 |
| | | Qwen3-VL-Plus | **68.02** | 51.84 |
| | | GLM-4.6V-Flash | **49.85** | 42.05 |
| Transient Response | CircuitSense | Gemini-3-Flash | 3.10 | - |
| | | GLM-4.6V-Flash | 1.02 | - |
| | AutoVSR | Gemini-3-Flash | **85.84** | 82.74 |
| | | GLM-4.6V-Flash | **63.25** | 62.23 |

**Comparison with Circuit-Specific CoT Strategies.** To verify the effectiveness of the AutoVSR framework, we

select CircuitSense as a specialized baseline method for circuit analysis. As shown in Table 3, on transfer-function generation, CircuitSense achieves accuracies ranging from 7.80% to 40.89%, while AutoVSR reaches accuracies from 49.85% to 82.85%. On transient response expression generation, CircuitSense achieves accuracies of 1.02% to 3.10%, while AutoVSR reaches 63.25% to 85.84%. This is because AutoVSR constructs symbolic equations through deterministic symbolic tools rather than intuitive model inference, ensuring that equation construction and algebraic computation are grounded in executable and verifiable intermediate results. In contrast, CircuitSense lacks a verifiable intermediate representation, causing errors from visual recognition or topology understanding to be directly propagated and amplified during symbolic derivation. For open source small scale models, such as GLM-4.6V-Flash, AutoVSR achieves accuracies of 49.85% on transfer function generation and 63.25% on transient response expression generation, while CircuitSense reaches only 7.80% and 1.02%. This result shows that AutoVSR is not limited to large scale models, but can also consistently perform well on models with constrained parameter sizes.

### 4.2. Ablation Study

Our ablation study investigates the contribution of different components in the AutoVSR framework. Specifically, we examine the effects of (i) disabling the task router, (ii) disabling the component rules library, (iii) disabling component detection, (iv) excluding the validation feedback flow, (v) disabling the solution planning module, and (vi) omitting the symbolic tools library, in comparison with the full AutoVSR framework. The results in Table 4 show that removing a single component consistently leads to a noticeable drop in accuracy. In general, the ablation results demonstrate that the performance gains of AutoVSR arise from the coordinated interaction of all modules, since removing components of the high or low-level consistently degrades reasoning accuracy.

### 4.3. Practical Applicability and Efficiency Analysis

To evaluate the practical applicability of AutoVSR, we apply it to lightweight VLM backbones including Gemini-3-Flash, Qwen3-VL-Plus, and GLM-4.6V-Flash, and compare the results with high-cost end-to-end models including Gemini-3-Pro (Team, 2025), GPT-5 (Singh et al., 2025), and Claude-Sonnet-4.5 (Anthropic, 2025b). We evaluate on 100 representative cases uniformly sampled from five circuit types, focusing on instances that require complex multi-step reasoning. AutoVSR achieves substantially higher accuracy ranging from 56.31% to 89.32%, while significantly reducing token consumption from 4,563 to 8,490 tokens and inference latency from 17.01 to 50.61 s. In comparison, the expensive models reach accuracies of 26.21% to 52.43%,

*Table 4.* **Ablation results of AutoVSR under different settings**, including different backbone models (Gemini-3-Flash and GLM-4.6V-Flash) and different ablated modules, namely the Task Router, Component Detection, Component Rules Library, Validation Flow, Solution Planning, and Tools Library. Overall, all modules contribute positively to the performance of AutoVSR.

| Model Variant | Ablation Settings | Accuracy (%) |
|---|---|---|
| AutoVSR w/ Gemini-3-Flash | w/o Task Router | 70.24 |
| | w/o Component Detection | 72.97 |
| | w/o Component Rules Library | 70.73 |
| | w/o Validation Flow | 74.97 |
| | w/o Solution Plan | 74.20 |
| | w/o Tools Library | 20.49 |
| | AutoVSR | **82.85** |
| AutoVSR w/ GLM-4.6V-Flash | w/o Task Router | 46.43 |
| | w/o Component Detection | 48.39 |
| | w/o Component Rules Library | 29.85 |
| | w/o Validation Flow | 31.31 |
| | w/o Solution Plan | 35.71 |
| | w/o Tools Library | 4.21 |
| | AutoVSR | **49.85** |

require 7,033 to 13,412 tokens, and inference latency from 89.52 to 148.4 s. This advantage stems from decoupling visual understanding from symbolic derivation through an executable intermediate representation and tool-based solvers, enabling lightweight models such as GLM-4.6V-Flash to outperform much larger end-to-end models. The detailed comparison across accuracy, token consumption, and inference latency is visualized in Table 5.

*Table 5.* **Comparison of Accuracy, Token Usage, and Runtime.** AutoVSR with lightweight VLM backbones achieves higher accuracy with fewer tokens and lower inference runtime than frontier models such as GPT-5, Claude Sonnet-4.5 and Gemini-3-Pro, demonstrating its practical efficiency for circuit symbolic expression generation.

| Method | Model | Acc. (%) | Token Usage | Runtime (s) |
|---|---|---|---|---|
| AutoVSR w/ Lightweight Models | Gemini-3-Flash | **89.3** | **4,563** | **17.0** |
| | Qwen3-VL-Plus | 74.8 | 5,487 | 35.5 |
| | GLM-4.6V-Flash | 56.3 | 8,490 | 50.6 |
| E2E w/ Frontier Models | Gemini-3-Pro | 52.4 | 13,412 | 118.0 |
| | GPT-5 | 45.6 | 7,152 | 148.4 |
| | Claude-Sonnet-4.5 | 26.2 | 7,033 | 89.5 |

## 5. Conclusion

This work presents AutoVSR, a visual-to-symbolic reasoning framework that enables symbolic analysis from visual inputs and task-specific queries using VLMs. Starting from visual observations and problem queries, AutoVSR constructs and verifies an executable intermediate representation, and then performs symbolic reasoning using tool-based solvers for equation construction and algebraic computation.

This approach overcomes the limitations of native reasoning models that rely on implicit inference, and enables reliable, interpretable, and efficient visual-to-symbolic reasoning. Experiments across diverse circuit benchmarks demonstrate that AutoVSR consistently outperforms native VLMs and specialized methods in accuracy, while significantly reducing inference cost and latency, showing strong potential for scalable and cost-effective visual-symbolic reasoning.

## Acknowledgements

This work is supported by the National Natural Science Foundation of China, under grant 92473115, U25A20447, and 62571184.

## Impact Statement

The proposed AutoVSR framework advances visual-to-symbolic reasoning for circuit understanding by enabling reliable symbolic expression generation directly from schematic images. Our approach addresses key challenges in this setting, including accurate visual-to-symbolic construction and correctness in multi-step symbolic derivation, which are critical for trustworthy circuit reasoning. By overcoming the limitations of end-to-end vision-language models that lack explicit verification and tool-level execution, AutoVSR enables accurate and verifiable symbolic expression generation while reducing error propagation. This not only improves the practicality of AI-assisted circuit reasoning but also provides a scalable framework that generalizes across diverse circuit types and analytical tasks. Our results demonstrate consistent performance gains over existing baselines, paving the way for future research on tool-augmented visual-to-symbolic reasoning in circuit analysis and related domains.

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

# A. Appendix

## A.1. Netlist Pipeline Example

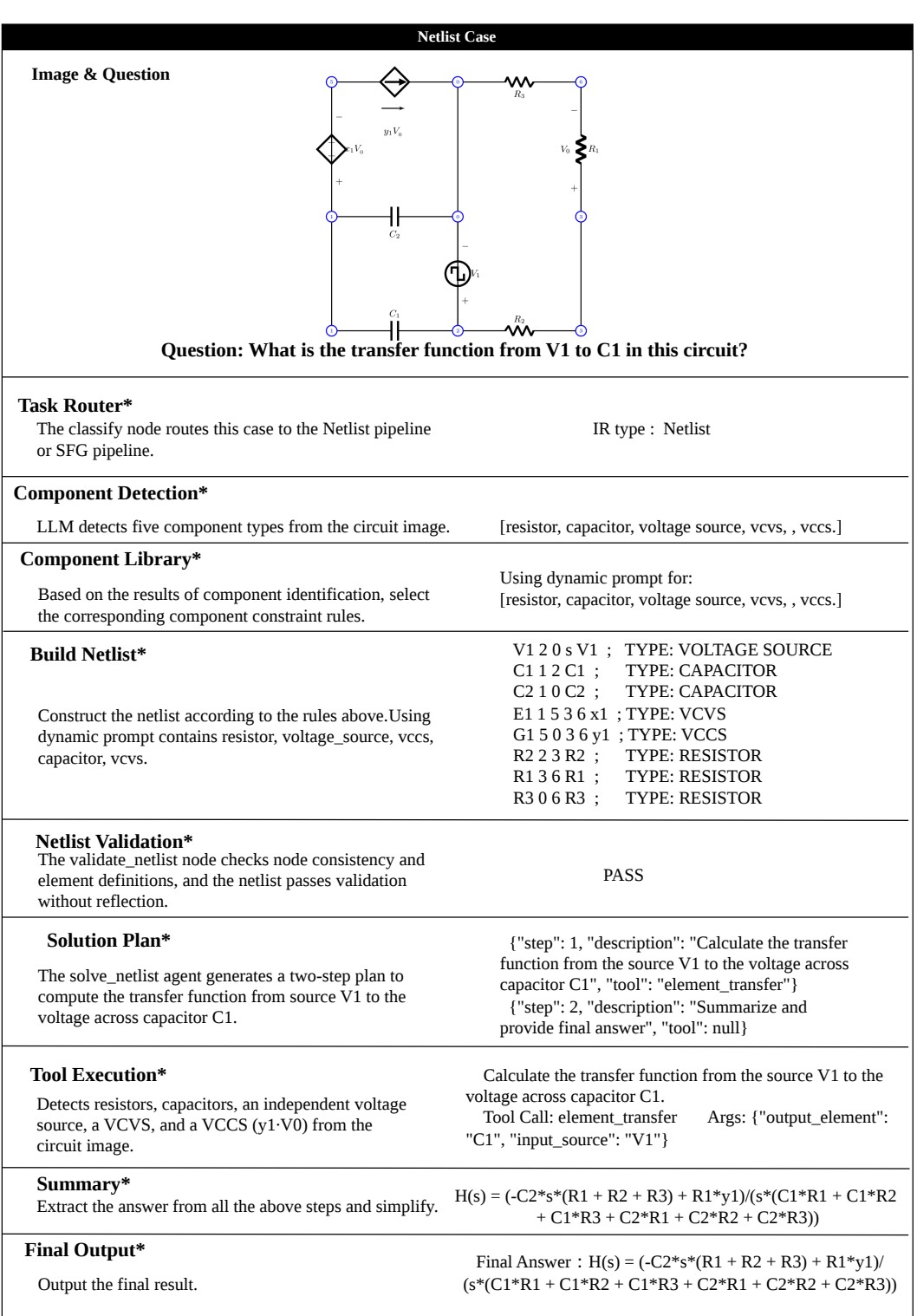

| Netlist Case |
|---|
| **Image & Question** |
| Question: What is the transfer function from V1 to C1 in this circuit? |

**Task Router***

The classify node routes this case to the Netlist pipeline or SFG pipeline.

IR type : Netlist

**Component Detection***

LLM detects five component types from the circuit image.

[resistor, capacitor, voltage source, vcvs, , vccs.]

**Component Library***

Based on the results of component identification, select the corresponding component constraint rules.

Using dynamic prompt for:
[resistor, capacitor, voltage source, vcvs, , vccs.]

**Build Netlist***

Construct the netlist according to the rules above. Using dynamic prompt contains resistor, voltage_source, vccs, capacitor, vcvs.

V1 2 0 s V1 ;  TYPE: VOLTAGE SOURCE
C1 1 2 C1 ;     TYPE: CAPACITOR
C2 1 0 C2 ;     TYPE: CAPACITOR
E1 1 5 3 6 x1 ; TYPE: VCVS
G1 5 0 3 6 y1 ; TYPE: VCCS
R2 2 3 R2 ;     TYPE: RESISTOR
R1 3 6 R1 ;     TYPE: RESISTOR
R3 0 6 R3 ;     TYPE: RESISTOR

**Netlist Validation***
The validate_netlist node checks node consistency and element definitions, and the netlist passes validation without reflection.

PASS

**Solution Plan***

The solve_netlist agent generates a two-step plan to compute the transfer function from source V1 to the voltage across capacitor C1.

{"step": 1, "description": "Calculate the transfer function from the source V1 to the voltage across capacitor C1", "tool": "element_transfer"}
 {"step": 2, "description": "Summarize and provide final answer", "tool": null}

**Tool Execution***

Detects resistors, capacitors, an independent voltage source, a VCVS, and a VCCS (y1·V0) from the circuit image.

Calculate the transfer function from the source V1 to the voltage across capacitor C1.
 Tool Call: element_transfer      Args: {"output_element": "C1", "input_source": "V1"}

**Summary***
Extract the answer from all the above steps and simplify.

H(s) = (-C2*s*(R1 + R2 + R3) + R1*y1)/(s*(C1*R1 + C1*R2 + C1*R3 + C2*R1 + C2*R2 + C2*R3))

**Final Output***

Output the final result.

Final Answer：H(s) = (-C2*s*(R1 + R2 + R3) + R1*y1)/ (s*(C1*R1 + C1*R2 + C1*R3 + C2*R1 + C2*R2 + C2*R3))

## A.2. SFG Pipeline Example

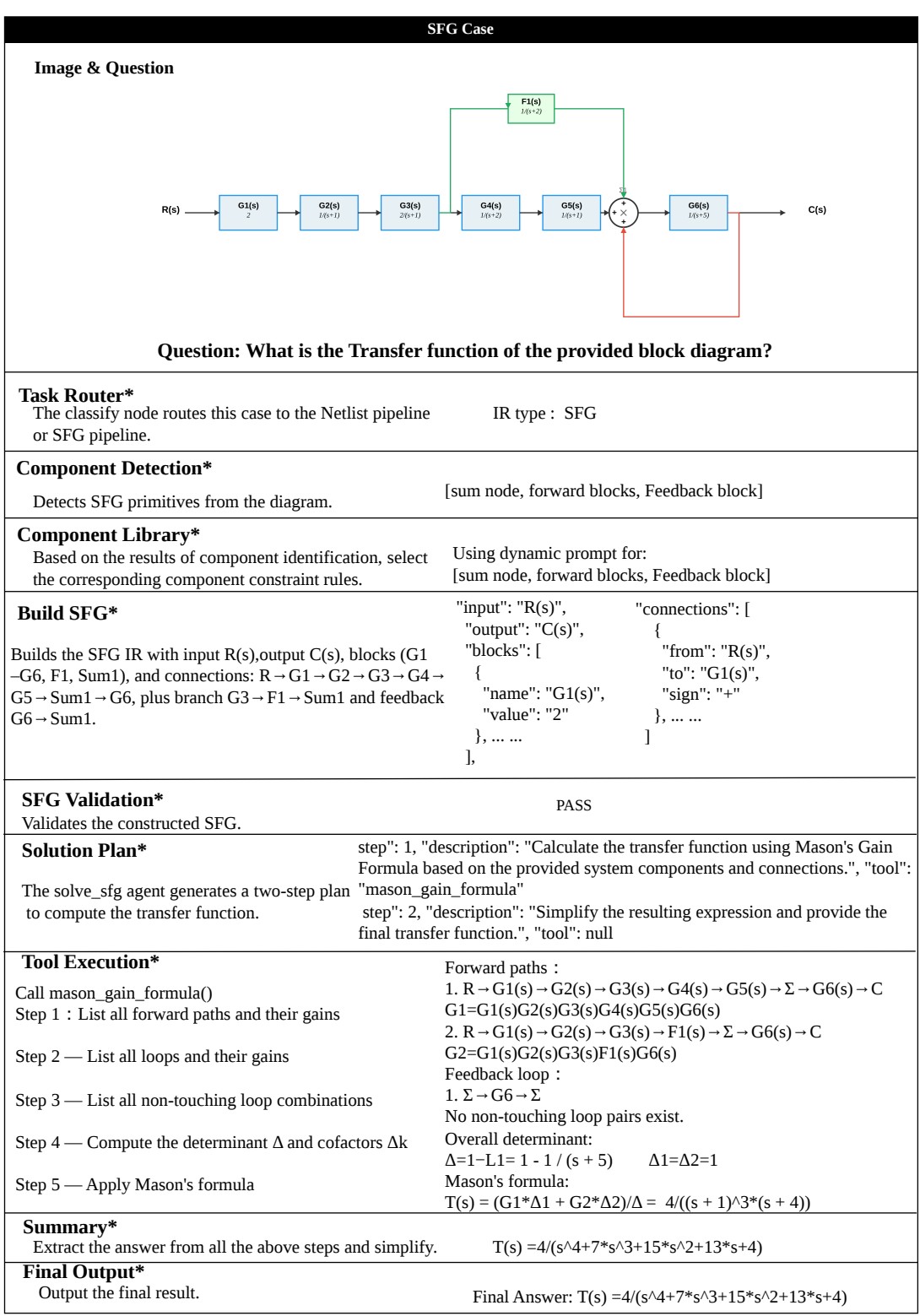

## A.3. Prompt Engineering Details

### A.3.1. TASK ROUTER

---

**Task Router (Classification Prompt)**

Classify the diagram type and extract analysis information.

**IR Type**:

**SFG** - Signal Flow Graph Block diagrams with transfer function blocks G(s), H(s) and summing junctions.

**Netlist** - Circuit Analysis(Lcapy) Linear circuits with R, L, C, sources, and controlled sources. Use for: transfer function, voltage gain, impedance, frequency response.

**Decision Rule**:

- Block diagram or signal flow graph - sfg

- Circuit schematic - netlist

- If unclear - netlist

**Analysis Type**:

- "transfer function": H(s), Vout/Vin

- "ac": Frequency response

- "dc": DC operating point

- "tran": Transient response

**Constraints**: Extract component values from the diagram or question:

- R, L, C values (e.g., "R1=10k, C1=1pF")

- Source values (e.g., "Vin=1V")

- Controlled source parameters (e.g., "gm=1mS")

**Output**:

"ir_type": "...", "analysis_type": "...", "constraints": "..."

Output ONLY JSON.

---

### A.3.2. COMPONENT DETECTION

---

**Component Detection Prompt (Netlist)**

Identify ALL component types present in this circuit schematic.

Look carefully at the image and list ONLY the component types you see:

**Component Types**:

- "resistor": R symbols, zigzag lines, rectangular boxes with R label

- "capacitor": C symbols, parallel plates, or labeled C values

- "inductor": L symbols, coils, or labeled L values

- "voltage_source": V symbols, circles with +/-, battery symbols

- "current_source": I symbols, circles with arrows

---

- "vcvs": Diamond voltage source controlled by voltage (E type)

- "vccs": Diamond current source controlled by voltage (G/gm type)

- "cccs": Diamond current source controlled by current (F type)

- "ccvs": Diamond current source controlled by current (H type)

- "opamp": Triangle with + and - inputs

- "conductance": G symbols as 2-terminal elements

**Output Format**: "detected_components": ["resistor", "capacitor", ...]

List ONLY the types that appear in the image.

Output ONLY JSON.

## Component Detection Prompt (SFG)

Identify ALL component types present in this signal flow graph.

Look carefully at the diagram and list ONLY the component types you see:

**Component Types**:

- "transfer_block": Blocks labeled with transfer functions (e.g., G(s), H(s), F(s))

- "summing_junction": Circular nodes with + / - signs indicating signal summation

- "signal_source": Input nodes such as R(s) or reference signals

- "signal_output": Output nodes such as C(s)

- "branch_point": Nodes where a signal splits into multiple paths

- "feedback_path": Explicit feedback connections forming closed loops

**Output Format**:

"detected_components": ["transfer_block", "summing_junction", ...]

List ONLY the types that appear in the image. Output ONLY JSON.

A.3.3. BUILD EXECUTABLE IR

## Build Netlist

You are an expert in electronic circuit analysis.

1. General Syntax: component-name Np Nm [args...]

2. Component Formats: Rname Np Nm Value, Vname Np Nm s Vin, ...

3. Key Rules: Node 0 = Ground, Consecutive node numbers, Unique names.

4. Source Prefix Rules: V=Voltage, I=Current, E=VCVS, F=CCCS...

5. Active Device Modeling: Op-amp: VCVS, MOSFET/BJT: Small-signal model.

## Build SFG

You are an expert in control systems and block diagrams. Analyze the block diagram image and extract the structure in JSON format.

1. JSON Structure: "blocks": [...], "connections": [...], "input": "...", "output": "..."

2. Block Format: "name": "G1", "value": "10/(s+1)"

3. Connection Format: "from": "NodeA", "to": "NodeB", "sign": "+"

4. Edge Cases: Feedback loops must be explicit.

A.3.4. VALIDATION IR

## Netlist Validation Prompt

You are an expert in debugging electronic circuit netlists.

You can see BOTH the original circuit schematic AND the generated netlist.

Analyze the validation errors by comparing the netlist with the schematic.

**Key Principle**:

The error messages already contain specific "Fix:" suggestions. Your job is to:

1. Compare the netlist with the circuit schematic

2. Understand WHY the error occurred (missing connection? wrong topology?)

3. Apply the suggested fix from the error message

**Common Error Types**:

- Floating nodes: Nodes not connected to ground : Add resistor to ground

- Voltage source removal: Differential input nodes only connected via V source : Add bypass resistors

- Missing ground: No node 0 : Ensure at least one component connects to 0

- Topology mismatch: Netlist doesn't match schematic : Check node connections

**Your Task**:

1. Look at the circuit schematic carefully

2. Compare with the netlist - identify what's missing or wrong

3. Read the "Fix:" suggestions in error messages

4. Provide specific fixes that match the original circuit

Output a clear list of what needs to be fixed.

## SFG Validation Prompt

You are an expert in debugging signal flow graphs and block diagrams.

Analyze the SFG validation errors and provide specific fix suggestions.

**Common SFG Error Patterns**:

- Missing input/output nodes : Add nodes like "R" or "C"

- Invalid block expressions : Use SymPy format: s**2, 1/(s+1)

- No path from input to output : Check edge directions

- Disconnected blocks : Ensure all blocks are in the signal path

- Invalid signs : Use "+" or "-" for feedback

**Your Task**:

1. Identify the root cause

2. Provide specific, actionable fixes

3. Focus on critical issues first

Output a clear list of fixes.

### A.3.5. SOLUTION PLAN

**Planning Prompt**

Generate a concise problem-solving plan.

**Rules**:

- Each step: tool call (tool: "name") OR reasoning (tool: null)

- Keep it simple: 2-4 steps is usually enough

- LAST step MUST be: "Summarize and provide final answer" with tool: null

- **Complexity**:

For circuits with $> 8$ nodes or complex block diagrams, use a 2-step flow: calculate transfer_function or voltage_gain using numerical values for efficiency.

**CRITICAL**: Output ONLY the JSON array. NO explanations, NO reasoning, NO other text.

### A.3.6. SYMBOLIC TOOL EXECUTION

**Execution Prompt**

Execute the current step of the plan.

**Current Plan**: plan_display

**Current Step (step_num)**: step_description

**Tool to use**: tool_name

**Instructions**: - Focus ONLY on completing this specific step

- Call the appropriate tool with correct parameters

- If no tool needed, provide your analysis

**Output Format**:

- Symbolic expressions/Transfer functions: Python/SymPy format (NOT LaTeX!)

A.3.7. SUMMARY

---

**Summary Prompt**

EXTRACT the final answer from tool results above.

**STRICT RULES**:

1. If tool returned "H(s) = <expression>", your answer IS that expression (copy it exactly)

2. Do NOT re-derive, re-calculate, or simplify $\rightarrow$ the tool result IS the correct answer

3. "0" is a VALID answer if the tool explicitly returned 0 $\rightarrow$ do not avoid it

4. For MCQ: compare tool result with options, output the matching number (1-5)

**Output (ONE LINE ONLY - NO REASONING)**: FINAL ANSWER: <copy the tool result or option number>

---

## A.4. Tool Library

*Table 6.* Netlist Tools

| name | input | output | description |
|------|-------|--------|-------------|
| voltage_gain | n1p, n1m, n2p, n2m | $A_v$ | Computes voltage gain $V_{out}/V_{in}$ between two defined ports. |
| node_transfer | out_node, in_src | $H(s)$ | Computes transfer function $H(s) = V_{node}(s)/V_{in}(s)$. |
| element_transfer | out_elem, in_src | $H(s)$ | Computes transfer function $H(s) = V_{element}(s)/V_{in}(s)$. |
| poles_zeros | tf_expr (optional) | Roots | Extracts poles and zeros of a transfer function for stability analysis. |
| get_voltage | target | $V(x)$ | Retrieves symbolic voltage at a node or across an element. |
| get_current | element | $I(x)$ | Retrieves symbolic current through a two-terminal element. |
| calculate | expr, params | Numeric | Evaluates symbolic expressions with numeric parameter substitution. |
| describe_analysis | None | String | Reports the analysis method used (DC, AC, or transient). |
| circuit_properties | None | String | Returns circuit properties such as causality and analysis domain. |
| list_components | None | String | Lists all nodes and elements in the circuit netlist. |
| impedance | node1, node2 (opt.) | $Z(s)$ | Computes symbolic impedance between two nodes. |
| thevenin | node1, node2 (opt.) | $V_{th}, Z_{th}$ | Computes the Thevenin equivalent, including output resistance. |
| state_space | state_order (optional) | $A, B, C, D$ | Generates state-space matrices with automatic simplification. |
| output_equation | output | Expression | Computes the output equation $y = Cx + Du$ for a specified output. |
| twoport_params | n1p, n1m, n2p, n2m, model | Matrix | Computes two-port parameters (H, Z, Y, S, A, etc.). |
| nodal_analysis | None | Equations | Returns nodal KCL equations in matrix form ($Av = b$). |
| mesh_analysis | None | Equations | Returns mesh KVL equations in matrix form ($Ai = b$). |
| noise_voltage | node | $V/\sqrt{\text{Hz}}$ | Computes noise voltage spectral density at a node. |
| frequency_response | tf_expr (optional) | $H(j\omega)$ | Computes frequency response magnitude and phase. |

*Table 7.* SFG Tools

| name | input | output | description |
|------|-------|--------|-------------|
| mason_gain_formula | None | $T(s)$ (String) | Computes the total transfer function using Mason's Rule and returns the simplified symbolic expression. |
| find_forward_paths | None | String | Lists all forward paths from input node to output node, including each path's node sequence and gain product. |

| name | input | output | description |
|------|-------|--------|-------------|
| find_loops | None | String | Lists all feedback loops in the signal flow graph, including loop node sequences and loop gain products. |
| simplify_expression | expr | String | Simplifies a symbolic expression (e.g., rational cancellation and factor reduction) using SymPy. |
| solve | None | Dict (result) | Core execution routine that coordinates forward-path discovery, loop discovery, determinant computation, and the final Mason aggregation $T(s) = \sum_k P_k \Delta_k / \Delta$. |
| _compute_delta | loops | $\Delta$ (Expr) | Computes the graph determinant $\Delta = 1 - \sum L_i + \sum L_i L_j - \cdots$ (currently includes up to non-touching loop pairs). |
| _compute_delta_k | loops, path | $\Delta_k$ (Expr) | Computes the cofactor $\Delta_k$ by excluding all loops that touch the $k$-th forward path. |
| _normalize_gain | gain | String | Normalizes gain strings for parsing (e.g., converts ^ to **, inserts implicit multiplication such as 3s→3*s). |
| _parse_gain_product | expr_str | Expr | Parses gain expressions into SymPy symbolic objects with automatic symbol discovery. |

## A.5. Component Library

*Table 8.* Component Rules Library

| component | syntax | input / nodes | description |
|-----------|--------|---------------|-------------|
| **Passive Elements (R, L, C, G)** | | | |
| Resistor | Rname Np Nm R | Np, Nm | Two-terminal resistor with resistance value $R$. |
| Capacitor | Cname Np Nm C | Np, Nm | Two-terminal capacitor with capacitance $C$. Voltage polarity defined as $v_C = V(Np) - V(Nm)$. |
| Inductor | Lname Np Nm L | Np, Nm | Two-terminal inductor with inductance $L$. Positive current direction is defined from Np to Nm. |
| Conductance (2-node) | Gname Np Nm G | Np, Nm | Two-node conductance element with value $G$. Distinct from VCCS controlled source. |
| **Independent Sources (V, I)** | | | |
| Voltage Source | Vname Np Nm [type] value | Np, Nm | Independent voltage source. Np is positive terminal, Nm is negative terminal. |
| Current Source | Iname Np Nm [type] value | Np, Nm | Independent current source with arrow direction from Np to Nm. |
| Source Type: Laplace | s | – | Laplace-domain source for transfer function analysis. |
| Source Type: DC | dc | – | DC source for operating-point analysis. |
| Source Type: Step | step | – | Step input for transient response analysis. |
| Source Type: AC | ac | – | AC phasor source defined by magnitude and phase. |

| component | syntax | input / nodes | description |
|---|---|---|---|
| Source Type: Time-domain | `{f(t)}` | – | Explicit time-domain source defined as a function of time. |
| **Controlled Sources (E, G, F, H)** | | | |
| VCVS (E) | `Ename Np Nm Ncp Ncm gain` | Np, Nm, Ncp, Ncm | Voltage-controlled voltage source. Output controlled by $V(Ncp) - V(Ncm)$. |
| VCCS (G, 4-node) | `Gname Np Nm Ncp Ncm value` | Np, Nm, Ncp, Ncm | Voltage-controlled current source (4-node). |
| CCCS (F) | `Fname Np Nm controlName gain` | Np, Nm | Current-controlled current source. Control current sensed via a voltage source. |
| CCVS (H) | `Hname Np Nm controlName gain` | Np, Nm | Current-controlled voltage source. Control current sensed via a voltage source. |
| Current Sensing Rule | `Vsense N1 N2 0` | N1, N2 | Zero-voltage source inserted to sense branch current. |
| **Active Devices (Operational Amplifier Model)** | | | |
| Op-Amp (VCVS macro) | `Eint1 Nout 0 0 31 Ad 0` | Nout, 31 | Operational amplifier modeled as a VCVS with open-loop gain $A_d$ and zero output resistance. |
| Op-Amp Rule | – | – | Explicit VCVS macro must be used; the `opamp` keyword is not allowed. |
| **Connections and Ports (W, O, P)** | | | |
| Wire / Short | `Wname Np Nm` | Np, Nm | Ideal short connection. No value parameter allowed. |
| Open Connection | `Oname Np Nm` | Np, Nm | Explicit open circuit between two nodes. |
| Port | `Pname Np Nm` | Np, Nm | Defines an external port for I/O or multiport analysis. |
| Impedance Label Rule | `RZL 3 0 ZL` | – | Symbolic impedance labels must still use R/C/L/G prefixes. |

