# OpenReview forum: "AutoVSR: Automatic Visual-to-Symbolic Reasoning for Symbolic Expression Generation from Circuit Schematic"
_ICML.cc/2026/Conference — ICML 2026 regular_

### Official Review · Reviewer_pKXD · 2026-03-08

**Soundness:** 2
**Presentation:** 3
**Significance:** 3
**Originality:** 3
**Overall Recommendation:** 4
**Confidence:** 3

**Summary:**

This paper introduces AutoVSR, a two-stage agentic framework for complex symbolic reasoning over circuit diagrams. In the first stage, it converts a circuit image into an executable intermediate representation (IR) using component-rule retrieval and dynamic context prompting. In the second stage, it performs symbolic reasoning over the IR through a plan-and-execute pipeline. This stage uses a VLM to guide the reasoning process while invoking symbolic tools for execution.

**Compliance With Llm Reviewing Policy:**

Affirmed.

**Final Justification:**

My questions are resolved by the rebuttal and I have no more concerns.

**Key Questions For Authors:**

Please see the questions in the weakness part. I would like to raise the score if the authors could properly address my concerns.

**Limitations:**

Yes

**Strengths And Weaknesses:**

Strengths:
1. The paper presents an agentic framework for circuit symbolic reasoning with several well-motivated and novel components, including the grounding of visual inputs into an executable intermediate representation with verification, as well as VLM-based planning and execution supported by symbolic tools.
2. The experimental results show that AutoVSR consistently outperforms strong VLM baselines and a circuit-specific reasoning method.
3. The paper includes extensive ablation studies and qualitative case analyses, which help validate the contribution of the proposed design choices.

Weaknesses:
1. A possible concern is that, although Tables 1 and 2 show AutoVSR outperforming both general-purpose VLMs and the included circuit-specific baseline by a large margin, it is unclear whether the comparison set is sufficiently strong. In particular, the related work mentions other potentially relevant methods, and it would strengthen the empirical evaluation to clarify why these were not included or to compare against stronger baselines where feasible.
2. The evaluation is entirely built on a filtered subset of CircuitSense and focuses only on transfer-function generation. It is unclear how well the framework generalizes to other symbolic circuit tasks, or more diverse real-world data.

---

> ### Author Rebuttal · Authors · 2026-03-31
>
> **Q1: Insufficient baseline comparison, with potentially stronger or more relevant baselines not included.**
>
> A1: Thank you for your suggestion. To our knowledge, existing methods for generating symbolic expressions directly from circuit schematics are limited. As shown in the table below, symbolic toolchain-based methods such as Lcapy [1] and SymPy [2] can perform symbolic computation but cannot process schematic images directly. VLM-based methods [4, 5, 6] focus on topology generation, netlist recovery, or parameter optimization rather than symbolic expression generation. Only CircuitSense [3] supports symbolic expression generation from circuit images. Based on this, we select two types of baselines. The first is mainstream end-to-end VLMs, which measure the direct capability of general multimodal models. The second is CircuitSense, which serves as the most closely aligned specialized baseline.
>
> |Method|Circuit Gen.|Circuit Verification|MLLM Input|Symbolic Expression Gen.|Training Free|
> |-|-|-|-|-|-|
> |Lcapy [1]||||✓|✓|
> |SymPy [2]||||✓|✓|
> |CircuitSense [3]|||✓|✓|✓|
> |NetListify [4]|✓||✓|||
> |AnalogCoder [5]|✓||||✓|
> |LaMAGIC2 [6]|✓|||||
> |AutoVSR|✓|✓|✓|✓|✓|
>
>
>
>
> In addition, we also compare AutoVSR with advanced closed source models. We enable reasoning mode for all of them and provide detailed step by step prompts. As shown in the table below, AutoVSR still achieves better performance with lower inference cost. This result shows that the advantage of AutoVSR is not limited to comparisons with general VLMs or circuit specific baselines. It also remains effective when compared with closed source models that have stronger reasoning ability.
>
> | Method     | Backbone Model    | Accuracy (%) | Runtime (s) | Tokens    |
> | ---------- | ----------------- | ------------ | ----------- | --------- |
> | SOTA Model | Gemini-3-Pro      | 52.4%        | 118.0       | 13,412    |
> |            | GPT-5             | 45.6%        | 148.4       | 7,152     |
> |            | Claude-Sonnet-4.5 | 26.2%        | 89.5        | 7,033     |
> |            | Avg.              | 41.4%        | 118.6       | 9,199      |
> | AutoVSR    | Gemini-3-Flash    | **89.3%**    | **17.0**    | **4,563** |
> |            | Qwen3-VL-Plus     | 74.8%        | 35.5        | 5,487     |
> |            | GLM-4.6V-Flash    | 56.3%        | 50.6        | 8,490     |
> |            | Avg.              | **73.5%**    | **34.4**    | **6,180**  |
>
>
>
> [1] Hayes, M. Lcapy: symbolic linear circuit analysis with python. PeerJ Computer Science, 2022.
>
> [2] Meurer, A., and et al. Sympy: symbolic computing in python. PeerJ Computer Science, 2017.
>
> [3] Akbari, A., and et al. CircuitSense: A hierarchical circuit system benchmark bridging visual comprehension and symbolic reasoning in engineering design process, ICRL, 2025.
>
> [4] Huang, C.-Y., and et al. Netlistify: Transforming circuit schematics into netlists with deep learning. MLCAD, 2025.
>
> [5] Lai, Y., and et al. Analogcoder: analog circuit design via trainingfree code generation. AAAI, 2025.
>
> [6] Chang, C.-C., and et al. LaMAGIC2: Advanced circuit formulations for language model-based analog topology generation. ICML, 2025.
>
>
>
> **Q2: Generalization unclear beyond transfer function generation to other symbolic expression tasks.**
>
> A2: Thank you for your question. AutoVSR is designed as a general framework for symbolic generation from circuit schematics and is not limited to the transfer function task.
>
> To verify generalization, we further conducted experiments on the transient response task with 3,644 samples, which is larger than the transfer function subset, as detailed in the table below. The results show that AutoVSR significantly outperforms both the end-to-end baseline and the CoT baseline, with average accuracy improvements of 72.77% and 72.49%, respectively. These results further demonstrate the generalization ability of our framework across different circuit analysis tasks.
>
>
>
> | Method      | Model          | **Transient Response Acc. (%)** | Transfer Function Acc. (%) |
> | ----------- | -------------- | ------------------------------- | -------------------------- |
> | E2E         | Gemini-3-Flash | 2.85%                           | 23.40%                     |
> |             | GLM-4.6V-Flash | 0.69%                           | 7.34%                      |
> | CoT         | Gemini-3-Flash | 3.10%                           | 40.89%                     |
> |             | GLM-4.6V-Flash | 1.02%                           | 7.80%                      |
> | **AutoVSR** | Gemini-3-Flash | **85.84%**                      | **82.85%**                 |
> |             | GLM-4.6V-Flash | **63.25%**                      | **49.85%**                 |

---

> > ### Author Rebuttal · Reviewer_pKXD · 2026-04-03
> >
> > I thank the author's response. I have no more concerns.

---

> > > ### Author Response · Authors · 2026-04-04
> > >
> > > Dear reviewer pKXD,
> > >
> > > Thank you very much for following up on our rebuttal materials and for your thoughtful reassessment resulting in a positive rating. Your insightful suggestions have significantly improved our manuscript.
> > >
> > > Best, Authors

---

### Official Review · Reviewer_9ACg · 2026-03-12

**Soundness:** 2
**Presentation:** 3
**Significance:** 3
**Originality:** 2
**Overall Recommendation:** 3
**Confidence:** 2

**Summary:**

The paper presents framework for generating symbolic circuit expressions  from schematic images. It decomposes the task into two stages 1) VLM-based construction of an executable intermediate representation   with iterative verification feedback, followed by  2) LLM planned symbolic derivation delegated to deterministic tools . The paper is evaluated in a subset of problems from CircuitSense.

**Compliance With Llm Reviewing Policy:**

Affirmed.

**Final Justification:**

I still believe there is limited methodological novelty. The entire system is a hand-crafted pipeline of several prompt templates, rule libraries, and engineering glue. There is no learned components, no training, no transferable insight to a general ICML audience . I think this is paper would be a better fit in a different venue.

**Key Questions For Authors:**

see above

**Limitations:**

yes, see above

**Strengths And Weaknesses:**

Strengths

  - The decomposition into visual IR construction + deterministic symbolic tools is well-motivated.
  - The 4-stage verification pipeline   is thorough and the dual-check mechanism is a reasonable engineering design
  - Consistent improvements across multiple VLMs



  Weaknesses

  - There is limited methodological novelty. The entire system is a hand-crafted pipeline of  several prompt templates, rule libraries, and engineering glue. There is no learned components, no training, no transferable insight to a general ICML audience .

- Lack of stronger/fair baselines. AutoVSR gets up to 3 correction rounds with structured verification feedback. From my understanding,  baselines get a single pass. No baseline with iterative self-refinement or even basic output validation is included. The true value of the engineered pipeline over simple retry+verify is unknown.

- Single dataset, single task. All evaluation is transfer function generation on a filtered subset of CircuitSense. Why not the entire dataset?

- Ablation confirms tools do the heavy lifting. Removing the symbolic tools library collapses accuracy to near-baseline levels. Does this suggest the main contribution is   on using tools/formula not the surrounding pipeline? If we gives access to the same tools to the base VLMs and an attempt to verify will the performance increase?

---

> ### Author Rebuttal · Authors · 2026-03-31
>
> **Q1: Limited novelty as a hand-crafted pipeline without transferable insight.**
>
> A1: Thank you for your feedback. AutoVSR is a training-free, rule-driven framework for image-to-symbolic reasoning, designed specifically for generating symbolic expressions from circuit schematics. Symbolic expressions are fundamental to circuit analysis, as they explicitly describe input-output relationships and support reasoning about key properties such as stability and dynamic response. For example, the transfer function of a first-order RC low-pass filter, $H(s) = \frac{1}{1 + RCs}$, reveals from pole analysis that increasing capacitance $C$ lowers the cutoff frequency. Since the task takes schematic images as input, traditional symbolic tools such as SymPy [1] cannot directly process them, leaving the workflow heavily dependent on manual effort.
>
> To address this limitation, we build executable intermediate representation construction, iterative repair based on verification feedback, and a symbolic solving mechanism that separates planning from execution. Together, they form a problem decomposition strategy for reliable image-to-symbolic reasoning, recovering structured representations from visual inputs and enabling reliable symbolic reasoning on that basis.
>
> **Q2：Lack of stronger/fair baselines.**
>
> A2: Thank you for your suggestion. Our three-round retry applies only to the IR construction stage to repair syntactic and physical errors, not to repeat the entire reasoning process. To isolate the impact of validation, we conducted an evaluation using the following control group settings:
>
> |**Model**|**Method**|**OverallAcc. (%)**|
> |-|-|-|
> |**Gemini-3-Flash**|End-to-End|23.40|
> ||CircuitSense (CoT Baseline)|40.89|
> ||w/o Validation Flow|74.97|
> ||**AutoVSR**|**82.85**|
> |**GLM-4.6V-Flash**|End-to-End|7.34|
> ||CircuitSense (CoT Baseline)|7.80|
> ||w/o Validation Flow|31.31|
> ||**AutoVSR**|**49.85**|
>
>
> As shown in the table, the value of our pipeline is demonstrated from two aspects. First, even without the validation mechanism (`w/o Validation Flow`), AutoVSR still outperforms both the E2E and CircuitSense baselines, showing that the performance gain does not come merely from retries. Second, comparing `w/o Validation Flow` with the full AutoVSR pipeline shows that the structured validation mechanism brings significant additional improvement, confirming the effectiveness of our verification feedback design.
>
> **Q3: Why not the entire dataset?**
>
> A3: Thanks for the question. Within CircuitSense [2], the transfer function and transient response tasks are the most representative, accounting for the majority of the benchmark (**62.7%, 5020/8006**). Furthermore, they represent the highest task complexity. For instance, the best baseline results for these two tasks are only 13% and 38%, respectively. In contrast, the remaining categories are ones where existing models can already achieve high accuracy with simple reasoning steps.
>
> To further evaluate the generalization capability of our method, we additionally conducted experiments on the transient response task with 3,644 cases using Gemini-3-Flash and GLM-4.6V-Flash. As shown in table below, AutoVSR achieves clear improvement on both tasks.
>
> |Method|Model|**Transient Response Acc. (%)**|Transfer Function Acc. (%)|
> |-----------|--------------|-------------------------------|--------------------------|
> |E2E|Gemini-3-Flash|2.85%|23.40%|
> ||GLM-4.6V-Flash|0.69%|7.34%|
> |CoT|Gemini-3-Flash|3.10%|40.89%|
> ||GLM-4.6V-Flash|1.02%|7.80%|
> |**AutoVSR**|Gemini-3-Flash|**85.84%**|**82.85%**|
> ||GLM-4.6V-Flash|**63.25%**|**49.85%**|
>
> **Q4: Ablation suggests symbolic tools drive performance, questioning the contribution of the surrounding pipeline.**
>
> A4: Thank you for your comment. Our symbolic tools require an instantiated circuit object as input, which cannot be obtained directly from a schematic image. Instead, it depends on a correctly constructed and validated executable IR. This IR generation in turn relies on dynamic context prompting and a multi-level validation mechanism. These modules work together, and each is necessary. The ablation results in Table 3 confirm that removing any single module leads to an accuracy drop.
>
> Therefore, even if a VLM is given access to the same tool library, it still cannot solve the task directly, as it has no means to convert a schematic image into an instantiated circuit object. The core contribution of AutoVSR lies precisely in this conversion, constructing and validating an executable IR that bridges visual input and symbolic tool execution.
>
> [1] Meurer, A., and et al. Sympy: symbolic computing in python. PeerJ Computer Science, 2017.
>
> [2] Akbari, A., and et al. CircuitSense: A hierarchical circuit system benchmark bridging visual comprehension and symbolic reasoning in engineering design process, ICRL, 2025.

---

> > ### Author Rebuttal · Reviewer_9ACg · 2026-04-03
> >
> > I thank the authors for the response, I have no more concern.

---

> > > ### Author Response · Authors · 2026-04-04
> > >
> > > Dear Reviewer 9ACg,
> > >
> > > Thank you very much for your acknowledgement. We truly appreciate your thoughtful feedback.
> > >
> > > Regarding Q2, specifically the concern about "lack of stronger/fair baselines," we would like to further note that we have also compared our method with stronger baselines in Section 4.3 and Figure 4 of Appendix 6. The detailed results are provided in the table below. As shown, AutoVSR with a lightweight model still outperforms commercial SOTA reasoning models, achieving an average accuracy improvement of 32.1% while using 3,019 fewer tokens and reducing runtime by 84.2 seconds on average.
> > >
> > > | Method     | Backbone Model    | Accuracy (%) | Runtime (s) | Tokens    |
> > > | ---------- | ----------------- | ------------ | ----------- | --------- |
> > > | SOTA Model | Gemini-3-Pro      | 52.4%        | 118.0       | 13,412    |
> > > |            | GPT-5             | 45.6%        | 148.4       | 7,152     |
> > > |            | Claude-Sonnet-4.5 | 26.2%        | 89.5        | 7,033     |
> > > |            | Avg.              | 41.4%        | 118.6       | 9,199     |
> > > | AutoVSR    | Gemini-3-Flash    | **89.3%**    | **17.0**    | **4,563** |
> > > |            | Qwen3-VL-Plus     | 74.8%        | 35.5        | 5,487     |
> > > |            | GLM-4.6V-Flash    | 56.3%        | 50.6        | 8,490     |
> > > |            | Avg.              | **73.5%**    | **34.4**    | **6,180** |
> > >
> > > If you have any further questions, please let us know. Otherwise, if you find our responses helpful, we would appreciate your consideration in updating score.
> > >
> > > Best regards,
> > > Authors

---

### Official Review · Reviewer_miDQ · 2026-03-13

**Soundness:** 3
**Presentation:** 3
**Significance:** 2
**Originality:** 2
**Overall Recommendation:** 4
**Confidence:** 3

**Summary:**

This paper engineers a vision-language question answering system on a circuit diagram.
The VLM looks at a circuit diagram and convert it into a netlist representation or a signal-flow-graph representation.
The generated IR is verified with a symbolic solver and the system performs a rejection sampling based on the result.
Finally, it produces the summary of the IR, from which the final answer is generated by the LLM.
The proposed approach significantly outperforms a naive VLM and a prior work on one of the task.

The approach proved to be another win for neuro-symbolic methods.
However, this paper is heavily application-oriented and it would be difficult to justify acceptance in ICML.
The paper goes into too many specific details of electric circuits, which makes it more appropriate to be submitted to a VLSI conference, etc.

**Compliance With Llm Reviewing Policy:**

Affirmed.

**Final Justification:**

the authors resolved the concerns.

**Key Questions For Authors:**

Runtime of CoT? What is the rejection rate? Did you try constrained decoding?

**Limitations:**

I do not find any specific limitations under the assumed target domain.

**Strengths And Weaknesses:**

strength: strong performance.

weakness:

Novelty: The system is basically a large rejection sampling loop with a symbolic verifier, which by itself has little novelty.
It also utilize the recently popular "agentic tool-calls" that allow llms to decide which verifier to use next.

Field mismatch: I believe it did nothing wrong, but perhaps this is slightly out of the topic in this conference due to its engineering-heavy / applicaton-oriented aspect. Chasing the state of the art is important, but there should be a better paper that could be accepted in this conference.

For an empirical paper, the paper surprisingly lacks any analysis of the system's runtime characteritics at all. How long does it take to produce an answer? What is the rejection rate? while the paper briefly discusses it, it does not compare them with e.g. CoT approach.

The authors mention that the IR generation also relies on syntax checker as on of the verifier. I wonder if it is much more efficient to do it instead with constrained / structured generation. Some of the symbolic rules may also be encoded in the CFG for constrained generation engines. Compare the runtimes with different method.

---

> ### Author Rebuttal · Authors · 2026-03-31
>
> **Q1: Novelty concerns over rejection sampling and agentic tool-calls.**
>
> A1: Thanks for the comments. Our method is not a rejection sampling strategy. Rejection sampling discards a failed candidate and starts over. Instead, our framework iteratively repairs the intermediate representation (IR) through four stages as shown in Figure 3, including generation, dynamic and static validation, error feedback, and targeted correction. Through our iterative repair mechanism, the final accuracy is improved by approximately 13.2% on average, according to Table 1 and 3.
>
> Secondly, our system differs from general agentic tool-calling, where a VLM reasons end-to-end over raw multimodal input and autonomously decides tool usage. Instead, we first convert the circuit schematic into a verified IR and load it as a circuit object. The solving process is then jointly constrained by the circuit structure encoded in the IR and the symbolic problem formulation, where the LLM handles solution planning using its circuit domain knowledge while symbolic tools execute the step-by-step computation. This separation improves the quality of the final symbolic expressions.
>
> **Q2: Field mismatch.**
>
> A2: Our work is consistent with the application driven machine learning track. Symbolic expression generation from circuit schematics addresses a real-world need. These expressions explicitly characterize input-output relationships and support reasoning about key properties, such as static and dynamic response, for circuit analysis.
>
> For example, the transfer function of a first-order RC low-pass filter, $H(s) = \frac{1}{1 + RCs}$, reveals directly from pole analysis that increasing capacitance $C$ lowers the cutoff frequency. Despite the availability of powerful symbolic tools such as Lcapy [1], none can directly process schematic images, leaving this workflow heavily dependent on manual effort in practice.
>
> The ML community has shown growing interest in circuit design area. Recent examples include AnalogCoder [2] for training-free analog circuit design, LaMAGIC2 [3] for circuit topology generation.
>
> Our work addresses the general problem of image-to-symbolic reasoning, where the goal is to extract reliable structured representations from visual inputs and perform symbolic reasoning on that basis. To tackle the key challenges of cross-modal understanding, structured representation construction, and multi-step symbolic reasoning, we propose executable IR construction, iterative repair via verification feedback, and a symbolic solving mechanism that separates planning from execution.
>
> **Q3: Missing runtime analysis and comparison with CoT baseline.**
>
> A3: Thanks for the suggestion. We provide a detailed runtime evaluation in Section 4.3 and Figure 4 of Appendix 6. The specific results are shown in the table below. Baselines use commercial SOTA models with their highest-level CoT enabled. AutoVSR uses lightweight models without relying on the models' internal CoT capability.
>
> The results show that AutoVSR achieves an average accuracy gain of **32.1%**, while using **3,019** fewer tokens and reducing runtime by **84.2 seconds** on average.
> |Method|Model|Accuracy (%)|Runtime (s)|Tokens|
> |:-|:-|:-|:-|:-|
> |SOTA Model|Gemini-3-Pro|52.4%|118.0|13,412|
> ||GPT-5|45.6%|148.4|7,152|
> ||Claude-Sonnet-4.5|26.2%|89.5|7,033|
> |**Avg.**|-|**41.4%**|**118.6**|**9,199** |
> |AutoVSR|Gemini-3-Flash|89.3%|17.0|4,563|
> ||Qwen3-VL-Plus|74.8%|35.5|5,487|
> ||GLM-4.6V-Flash|56.3%|50.6|8,490|
> |**Avg.**|-|**73.5%**|**34.4**|**6,180**|
>
> **Q4: Missing comparison with constrained/structured generation.**
>
> A4: Thanks for the advice. We agree that constrained or structured generation can help reduce syntactic errors. However, our IR serves as a representation, a verifiable structure, and an execution interface. The example is as follow:
> ~~~
> V1 4 0 step
> Rint1 4 31 Rint1
> Cint1 31 3 Cint1
> Eint1 3 0 0 31 Ad
> R3 3 0 R3
> ~~~
> Representation: The IR explicitly encodes component types and node connectivity. For example, `R3 3 0 R3` indicates resistor R3 is connected between node 3 and ground node 0.
>
> Verification: Once loaded, the system checks whether the MNA matrix is invertible. For instance, node 31 is connected to the rest of the circuit only through `Rint1` and `Cint1`, so it must be verified that this node does not become a floating node.
>
> Execution: After verification, Lcapy instantiates the IR as a circuit object, constructs symbolic equations centered on the VCVS (`Eint1`).
>
> In contrast, CFG methods must convert the representation into SPICE-like syntax before invoking any symbolic solver, making the overall pipeline less efficient than ours.
>
> [1] Hayes, M. Lcapy: symbolic linear circuit analysis with python. PeerJ Computer Science, 2022.
>
> [2] Lai, Y., and et al. Analogcoder: analog circuit design via trainingfree code generation. AAAI, 2025.
>
> [3] Chang, C.-C., and et al. LaMAGIC2: Advanced circuit formulations for language model-based analog topology generation. ICML, 2025.

---

> > ### Author Rebuttal · Reviewer_miDQ · 2026-04-06
> >
> > Thanks for the replies.
> >
> > I am sorry that I missed the primary area designation as applications. As an application paper this is probably an ok paper, although I usually do not get overly excited by them. I always pick a fundamental research paper over an application paper.
> >
> > It is correct to say a repair loop is different from rejection loop, but the point still stands --- everyone does the same thing in code generation --- using compilers or type checkers as verifiers. I acknowledge however that perhaps there is a novelty in applying the same scheme in circuits, and proposing an IR.
> >
> > You could highlight the runtime & tokens more, since it is probably the largest concern of the actual users. You could also provide an estimated cost in dollars --- "we can do this in $$, while previous work required $$$."
> >
> > I should probably raise the score a bit.

---

> > > ### Author Response · Authors · 2026-04-06
> > >
> > > Dear Reviewer miDQ,
> > >
> > > Thank you very much for your acknowledgement and thoughtful suggestions. We greatly appreciate your feedback and will carefully summarize and incorporate your comments into the revised manuscript. Your valuable advice will substantially strengthen and improve our manuscript.
> > >
> > > Best regards,
> > > Authors

---

### Decision · Program_Chairs · 2026-04-30

**Decision:**

Accept (regular)

**Comment:**

The paper tackles the challenge of generating accurate symbolic mathematical expressions directly from circuit schematic images. Because end-to-end VLM based methods frequently hallucinate or fail at strict multi-step mathematical derivations, the authors introduce AutoVSR, a decoupled neuro-symbolic framework. First, the system translates the visual circuit into an "Executable Intermediate Representation" (IR), utilizing dynamic rule retrieval and a rigorous verification-feedback loop to catch topological and syntactic errors. Once the circuit is accurately grounded in text, an LLM acts as a reasoning planner, delegating the actual algebraic derivations to deterministic symbolic tools (like SymPy and Lcapy) to guarantee mathematical correctness.

Reviewers praised the framework's strong empirical performance but raised several methodological concerns. Reviewers pointed out that the system operates as a heavily engineered pipeline without newly learned components, questioning its fundamental novelty and generalizability. Furthermore, the reviewers requested comparisons against stronger frontier models, ablation studies to ensure the performance wasn't merely an artifact of "retry" mechanisms, and broader testing beyond the initial subset of transfer function problems.

In their rebuttal, the authors delivered highly compelling new evidence that successfully resolved these concerns. They expanded their evaluation to include a Transient Response task with over 3,600 samples, demonstrating massive performance gains and proving the framework's generalizability. Crucially, they benchmarked AutoVSR (powered by lightweight, open-weight models) against frontier models like GPT-5 and Gemini-3-Pro. AutoVSR achieved over 30% higher accuracy while consuming significantly fewer tokens and operating up to an order-of-magnitude faster. While the work relies heavily on prompt engineering and tool integration, the resulting empirical leap in a highly complex application domain justifies acceptance for the applications area.